# OpenReview forum: "SteeringSafety: Benchmarking Representation Steering in LLMs Across Safety Perspectives"
_ICML.cc/2026/Conference — ICML 2026 regular_

### Official Review · Reviewer_j7Gy · 2026-03-03

**Soundness:** 1
**Presentation:** 2
**Significance:** 2
**Originality:** 3
**Overall Recommendation:** 3
**Confidence:** 4

**Summary:**

The paper proposes a benchmark to study the effectiveness of representation steering in models while being aware of entanglement: where steering a certain concept/property may also alter other properties of the model. They perform this study on steering for harmfulness, bias mitigation, and hallucination mitigation, and observe entanglement effects across a variety of alignment and reasoning properties of the model.

**Compliance With Llm Reviewing Policy:**

Affirmed.

**Final Justification:**

Please refer to the below rebuttal/rebuttal reply discussion: there are a number of terminological, presentation, and explanatory flaws in the paper as it stands. Therefore, there appear to be too many required changes for me to certify without looking at a revision of the paper itself, which is not possible for this venue.

My concerns and final score justification are therefore, summarized in the following words, as in my last response to the authors (to their rebuttal reply).

> As it stands, it is unfortunate that I cannot see a revision of the paper here, given the scope and number of changes proposed. Therefore, I won't be able to certify this paper as a fully finished product. However, I shall recognize that the authors have addressed a number of my concerns, even if not all of them (especially not being able to see the paper with significant revisions) and have engaged with my concerns well and eagerly.

> I shall raise my score to a weak reject, to account for both my remaining concerns/hesitation but also to reflect the concerns that have been addressed.

That said, I do encourage the authors to improve their work, regardless of acceptance to this venue, as with the required corrections (perhaps more, wherever needed) it can become a full-fledged and helpful scientific contribution. At this moment, it does not meet the mark fully, but the rebuttal process has been helpful and is a good start.

**Key Questions For Authors:**

- [Q1] “Negative completions are generated using GPT-4.1-mini for the unanswerable set and randomly chosen where they already exist in other datasets.” Please clarify the random selection method. What are the system prompts used for GPT-4.1-mini (and other models in other evaluations in this paper)? What do the faitheval subset types mean?
- [Q2] Refer to W2 in weaknesses.
- [Q3] Refer to W3 in weaknesses.
- [Q4] Address weaknesses W1 and W1.1, and please provide a clear discussion about each steering type and in the results section connect the results discussing each steering type's observed behavior to the way it works/is defined.
- [Q5] Address W6 and W7 and any ambiguities therein.
- [Q6] Can you substantiate Finding 1 (refer to weakness W9) by comparing harmfulness steering effectiveness vs. safety steering effectiveness w.r.t. the *same* notions of safety? Else this claim is at worst wrong, and at best wholly unsubstantiated.
- [Q7] Define missing terms in W11; however, unlike the importance of describing the steering methods in the main text, it is sufficient to add these missing terms' definitions (in W11) to the appendix.
- [Q8] “removing settings where average KL divergence on last-token probabilities is less than 0.1 (Arditi et al., 2024)” What is a setting here, precisely? What does it comprise of? Is it a tuple of layer index, steering method, prompt, etc.? A line or two of explanation, especially by condensing space consuming figures, would better aid the presentation of the paper.
- [Q9] Address weaknesses W13 and W13.1

**Limitations:**

Yes, they have to some extent. However, some of their takeaways are not fully substantiated and a key limitation (not exhaustive) is that their harmfulness steering tests are not contrasted against a harmfulness *reduction* steering test in the *same* domains for studying the difference in effectiveness between these kinds of steering.

**Strengths And Weaknesses:**

## Strengths

- [S1] Tackles a timely problem with actionable insights for practitioners on how to audit models for entanglement in and effectiveness of steering. Provides a useful suite of evaluations and benchmarks to perform this study over.
- [S2] In particular, insights on how this audit may be performed on a smaller model from the same family as the target model (which is to be steered) is very convenient and helpful.
- [S3] Indeed, the metric that is the ratio of effectiveness score to entanglement provides a succinct and elegant way of viewing a model's steerability without unintended effects in a single scalar.
---
## Weaknesses

Unfortunately, the current version of the paper, while attempting to analyze an important problem, lacks soundness of analysis, has significant issues with presentation, defining key concepts, unsubstantiated takeaways, etc.

- [W1] Several terms in the paper are left undefined or are poorly and ambiguously defined. The most concerning is how, while this paper primarily (even purely) studies them, the steering methods are not defined in the paper, with the authors only providing their abbreviations and associated citations. Refer to lines 226 - 238: can you please briefly explain each of these methods in the main text? If the paper is meant to study steering methods in depth and provide scientific insights about them, it is *essential* to put the descriptions about these methods front and center, even if they are short condensed descriptions. Else the main text is incomplete and the presentation lacks rigor as for a venue like ICML, nor will a reader be able to understand why certain methods behave the way they do. Again, it is highly preferable to prioritize such descriptions over whitespace or figures with excessive amounts of empty space. This makes it hard to study and appreciate the results properly.
   - [W1.1] Indeed, several discussions in the results section corresponding to discussing and comparing steering methods' impacts, the discussion is done by only talking about the results at face value without providing actual insights about "why" they may be occurring (as in why a particular method under/overperforms relative to another method), which is essential for a scientific study. For instance, look at the discussion in 4.0.4, which merely describes the results at face level without providing any useful or actionable insights (for instance, variant X outperforms Y because of this property…).
- [W2] “Sneaking uses adversarial DarkBench (Kran et al., 2025) prompts to test if the model subtly shifts the original stance when reframing opinions.” What is sneaking? This is not well-defined in the paper.
- [W3] “Political Views uses prompts from TwinViews-13k (Fulay et al., 2024), which ask the model to agree with either left or rightleaning opinions. We report the percentage of responses choosing the left-leaning option since models often skew left (Fulay et al., 2024; Potter et al., 2024). Unlike other datasets where higher is better, this convention was chosen arbitrarily.” This seems a bit hand-wavey and speculative and may not lead to a sound evaluation. Why not define the metric as simply the absolute value of the difference of the percentage of responses that lean each way?
- [W4] “We include adversarial refusal ablation” The term "adversarial refusal ablation" is left undefined.
- [W5] Indeed, in light of all these missing discussions and definitions, the empty space in Figure 1 (which takes up a whole page without providing any useful scientific insight and can be significantly compressed losslessly!), Figure 3, and the white space on page 7 seem highly unnecessary and can be removed/reduced in favor of including these discussions.
- [W6] “Entanglement is not normalized as there is often minor entanglement across all steering methods, which may show a large relative increase but not be meaningful.” This is ill-phrased and unclear. What a large relative increase in what quantity and relative to what? Relative to steering methods? Relative to other alignment desiderata? Similarly, normalized w.r.t. what? This argument for not normalizing entanglement is not informative or convincing.
- [W7] And if there is minor entanglement, why is there large increase? And what do you mean by meaningful exactly? Show an example where a large entanglement value may not yield a meaningful (in whatever quantity the authors refer to) change.
- [W8] **Unclear if political leanings are due to/affected by steering at all** “While both achieve reductions in hallucination, entanglement is inconsistent even in direction, with Gemma-2-2B becoming more left-leaning while Llama-3.1-8B becomes more right-leaning.” This would be better appreciated if it were to be shown what the initial leanings of each base model (unsteered) were. For instance, if a model were to be more left-leaning to begin with and didn’t actually change in its leaning after steering, its political stance cannot be attributed to a change caused by steering (because there was actually no change). Put another way, the way political leanings are reported in the context of steering in this paper does not show if there was any shift due to steering.
  - [W8.1] About the metric used: “Political Views uses prompts from TwinViews-13k (Fulay et al., 2024), which ask the model to agree with either left or rightleaning opinions. We report the percentage of responses choosing the left-leaning option since models often skew left (Fulay et al., 2024; Potter et al., 2024). Unlike other datasets where higher is better, this convention was chosen arbitrarily.” This seems a bit hand-wavey and may not lead to a sound evaluation. Why not define the metric as simply the absolute value of the difference of the percentage of responses that lean each way?
- [W9] **Important: Key claim in finding 1 is not substantiated** “This is concerning as it means it is much easier to decrease safety with the selected steering methods rather than increase it.” This is not clear and unsubstantiated; where are the effectiveness scores for safety steering to contrast this with? If you mean to substantiate it via “Finding 1: Across all models, it is easier to decrease safety via increasing harmfulness than it is to decrease hallucination and bias.” then it does *not* prove this point, actually. The notions of harmfulness and those of hallucination and bias may be weakly anti-correlated (for instance, if by harmfulness you mean something like toxicity or forbidden fraud or weapon development, then bias and hallucination mitigation is not the opposite of it, but steering to induce model refusals against such threats is). Also the effectiveness quantification for bias seems unclear and problematic, see below.
- [W10] “suggesting that either these models are already well-aligned on demographic bias or that current steering techniques struggle with more subtle behavioral modifications.” Can you check? Can you provide bias benchmark results for the base models to substantiate these speculations?
- [W11] In Figure 4, what do standard, conditional, and NoKL mean? Define them rigorously, the paper is missing these details. Again, to reiterate, it is key to discuss all steering methods and settings, which is missing from the main text and is a significant issue in terms of presentation. They are briefly mentioned in Sec 4.0.4 without a proper explanation. The paper should ideally have a description, even if in the appendix. Therefore, it is impossible to see or understand why each variant succeeds/fails. Indeed, even the discussion in 4.0.4 merely describes the results at face level without providing any useful or actionable insights (for instance, variant X outperforms Y because of this property…).
- [W12] Not a major weakness but this calls for taking a bit of a grain of salt and possible need for moderating a few claims: the strongest entanglement results come from harmfulness; using these to claim that helpful steering leads to substantial unintended effects is misleading as this kind of steering is *not* helpful in nature. This exaggerates the extent of unintended impacts from steering. However, the fact that unintended impacts (to whatever degree) do occur is valid, and sound.
- [W13] “Entanglement is not normalized as there is often minor entanglement across all steering methods, which may show a large relative increase but not be meaningful.” This is ill-phrased and unclear. What a large relative increase in what quantity and relative to what? Relative to steering methods? Relative to other alignment desiderata? Similarly, normalized w.r.t. what? This argument for not normalizing entanglement is not informative or convincing.
  - [W13.1] And if there is minor entanglement, why is there a large increase? And what do you mean by meaningful exactly? Show an example where a large entanglement value may not yield a meaningful (in whatever quantity the authors refer to) change.
---
These are significant, numerous, and grave flaws, and I encourage the authors to significantly improve the soundness and presentation of their paper, which otherwise attempts to tackle an interesting and important problem.

---

> ### Author Rebuttal · Authors · 2026-03-28
>
> We thank the reviewer for identifying several presentation improvements we will make in the revision. We note that several weaknesses appear to stem from misreadings of content present in the paper.
>
> ---
>
> **W1, W2, W4, W11, Q7, Q8: Undefined or missing terms and descriptions**
>
> **Clarification: Full method descriptions are in Appendix B; we will add brief inline summaries to the main text in the revision.**
>
> Lines 226--238 cite the originating paper for each method but do not describe how each works; full descriptions are in Appendix B.1.1. We will add brief prose summaries of each method's direction generation and application logic. "Adversarial refusal ablation" and DarkBench split names such as Sneaking (defined by Kran et al. (2025) as testing whether a model subtly shifts the original stance when reframing opinions) are terminology from cited prior work; we will replace or formally introduce each. Settings are Standard, NoKL, and Conditional, defined at lines 224 and 385--390; we will add forward references. The KL check discards directions where average KL divergence on last-token logits exceeds 0.1; NoKL removes this check; Conditional only applies steering when the cosine similarity between the direction vector and prompt activations exceeds a threshold, as defined in B.1.3. We will also resolve layout issues in the revision.
>
> ---
>
> **W1.1, Q4: Results at face value without mechanistic insight**
>
> **Clarification: Mechanistic attribution is intentionally scoped out as not yet tractable; the results section characterizes phenomena rather than explains them.**
>
> We agree causal explanations are a desirable goal. However, we argue it is presently not achievable. Mechanistic understanding of feature geometry has been investigated since Elhage et al. (2022), yet remains theoretical or applied only on small toy models, including extremely recent work post-dating our submission [1]. This is a deliberate scope boundary noted in Section 4.1: our goal is to establish what entanglement patterns exist and provide empirical foundations for future mechanistic work. We will make this framing more explicit at the start of the results section.
>
> ---
>
> **W3, W8, W8.1, W10: TwinViews metric unclear; unsteered baseline not shown**
>
> **Misunderstanding Clarified: All values are deltas relative to the unsteered baseline, shown at the top of every results figure.**
>
> All TwinViews and bias values are changes relative to the unsteered model's baseline, directly answering whether steering shifted behavior. Left-leaning responses are positive by convention but the metric is internally consistent with shifts in either direction; this only changes final reported signage. Unsteered baseline performance on all datasets is displayed in Figures 6-16. We will clarify the delta framing in the main text.
>
> ---
>
> **W6, W7, W13, W13.1: Entanglement normalization justification unclear**
>
> **Clarification: Normalization would create combinatorially incomparable values; absolute magnitude is more interpretable.**
>
> Normalizing by baseline headroom would make small absolute changes on high-baseline perspectives appear disproportionately large: a 1% absolute change on a 99% baseline perspective would appear equivalent to a 100% change on a 0% baseline perspective, despite having negligible practical impact. More fundamentally, each of our benchmarks has a different baseline for each of the models; normalized values would be incomparable across every model-dataset combination. Absolute changes allow readers to directly assess practical significance. We will replace the current passage with this explanation.
>
> ---
>
> **W9, Q6, W12: Key Finding 1 unsubstantiated; harmfulness steering framing**
>
> **Clarification: Safety-increasing steering degenerates to trivial blanket refusal; jailbreaking results are informative of representation geometry.**
>
> Arditi et al. (2024) and COSMIC (2025) document that applying the refusal vector in the safety-increasing direction causes indiscriminate refusal of all prompts, producing no meaningful signal throughout the evaluation. Additionally, as shown in Figures 6-16, baseline ASR is often already below 0.05. Steering toward zero is unrepresentative of practical settings and would likely exhibit extreme overfitting artifacts. We will cite this prior work directly and reframe results accordingly.
>
> ---
>
> **Q1: Negative completion selection; GPT-4.1-mini prompts; FaithEval subsets**
>
> Negative completions are drawn uniformly from existing incorrect answers in other datasets without replacement. Full LLM judge prompts are in Appendix E.1; we will also add the GPT-4.1-mini generation prompts. FaithEval subsets (Counterfactual, Inconsistent, Unanswerable) follow the taxonomy of Ming et al. (2025); we will add inline definitions in the main text.
>
> [1] Georgi Ivanov, Narmeen Oozeer, Shivam Raval, Tasana Pejovic, Shriyash Upadhyay, Amir Abdullah: Spectral Superposition: A Theory of Feature Geometry. CoRR abs/2602.02224 (2026)

---

> > ### Author Rebuttal · Reviewer_j7Gy · 2026-03-31
> >
> > Thank you to the authors for their response. Allow me to respond to the rebuttal below, with some points of agreement, disagreement, and the current verdict from me as a reviewer. Please note that the raised weaknesses are based on a careful reading of the paper over 2-3 in-depth fully-manual passes, and not mostly due to misunderstanding.
> >
> > ## Agreements
> >
> > W3: I take the authors’ point about the metric for political leaning from TwinPeaks and consider it resolved.
> > W6, W7: While I would appreciate further clarification on this point, in particular on the meaning of the phrase “a 1% absolute change on a 99% baseline perspective would appear equivalent to a 100% change on a 0% baseline perspective”, I will give the authors the benefit of the doubt and say this is largely resolved. Also, the authors need to be careful about how they write about a large increase, which may be perceived as a large increase in entanglement, and not in terms of the normalized quantity.
> > W8, W10: Thanks for pointing out the existence of the unsteered values in the appendix. However, this detail that the unsteered values are reported in the appendix is omitted from the main text, and this must be mentioned clearly in the revision.
> >
> > Comment: W13 and W13.1 in the review were redundant and repeat W6 and W7, and by that token, they also stand addressed  (apologies for this formatting error).
> >
> > ## Disagreements
> >
> > Regarding W9, I disagree with the authors majorly, and their point is refuted by recent published work. While steering for refusal may lead to overrefusal, recent published work (Zhao et al, NeurIPS 2025) show that harmfulness (ergo the harmlessness concept in the same domain) are encoded separately from refusal. So the authors’ response conflates refusal steering with harmlessness steering. This is a major flaw and remains unaddressed. The takeaway statement in the paper “This is concerning as it means it is much easier to decrease safety with the selected steering methods rather than increase it.” is therefore incorrect.
> >
> > As for Q8, the authors merely restated how they removed settings, which was already clear to me. My question had to do with how a setting is precisely defined, not the setting removal paradigm, and I would ask the authors to please reread my question, and I’d appreciate an updated response that answers the intended question to help my reassessment.
> >
> > Regarding W1.1, while full explanations remain implausible, but a qualitative, even speculative, assessment based on what each steering method’s methodological details may still be done. Right now, taking inferences at face value is not very scientifically informative or insightful, which is the very objective of such evaluation and benchmarking exercises. The authors may consider adding some plausible explanations for why a certain steering method outperforms another (does it rely on a particular quantity over what another underperforming method does? Such a discussion is important) (add citations if possible). Therefore, I would consider this concern as remaining partially unaddressed.
> >
> > W12 is also unaddressed: my point was that given that the strongest steering results presented in the paper correspond to steering to increase harmfulness, using these results to claim that helpful steering leads to substantial unintended effects is misleading. This part is not addressed.
> >
> > ## Neutral
> >
> > W2, W4, W11 (*Conditional*): I consider these *conditionally* resolved based on if the descriptions of this existing methods are included in a revision of the paper. However, as a whole, the presentation problems in the paper that were pointed out (and in general) are wide-ranging and this may require a major rewrite of the paper in this format and page limit, as opposed to minor inclusions or clarifications.
> >
> > **Verdict:** To sum up, this paper has some potential scientific merit and promise, and the authors addressed some outstanding concerns, but to be frank, the current product is simply not finished or fully sound and may benefit from technical improvement, corrections in some experiments’ design, and a *major* rewrite (and currently it appears rushed and imprecise) followed by a resubmission to a future venue. As it stands, the scope of the suggested improvements, both in terms of terminology and rewriting, is too massive for me to consider my concerns sufficiently addressed or recommend acceptance. However, I fully encourage the authors to improve their work and continue their investigation in the interest of making positive impact for the community.
> >
> > I also would like to note that the 5000 character limit for the rebuttal may not have allowed for fully descriptive responses so I would invite the authors to further engage with my concerns, and I remain happy to discuss further.

---

> > > ### Author Response · Authors · 2026-04-06
> > >
> > > We thank the reviewer for their continued and clearly structured engagement, and for the acknowledgment of our work's potential. As the reviewer notes, we are constrained to 5000 characters. In summary, we directly address the two primary substantive concerns (W9, W12) by tightening our framing and clarifying the scope of our claims, address remaining technical clarifications (e.g., Q8), and commit to incorporating additional definitions and intuition to improve presentation in the revision.
> > >
> > > ---
> > >
> > > **W12: Framing of harmfulness entanglement**
> > >
> > > We would like to clarify the scope of SteeringSafety's contributions. We do not claim to rigorously test **helpful** steering. SteeringSafety benchmarks existing steering methods as deployed in the literature, enabling a comparison of method-level behavioral differences across multiple safety perspectives when tested on consistent generation/evaluation data. Refusal suppression is the primary use case for which DIM and ACE were introduced (Arditi et al., 2024; Marshall et al., 2024); a benchmark restricted to helpful use cases would not represent the field as it exists. The entanglement we observe characterizes these methods as the field uses them (lines 168-177).
> > >
> > > ---
> > >
> > > **W9: Finding 1 and the harmfulness naming convention**
> > >
> > > We concede that the "harmfulness" label is a source of confusion. The label followed the naming convention of prior work which categorizes the contrastive datasets as harmful or benign. The intervention itself is refusal steering via orthogonalization of the refusal direction (Arditi et al., 2024; Marshall et al., 2024) and will be renamed. Within the directions studied, refusal suppression is substantially more effective than steering hallucination or bias. Key Finding 1 reports this directional asymmetry, specific to the perspectives evaluated. We will tighten this framing in the revision.
> > >
> > > ---
> > >
> > > **W1.1: Qualitative method-level explanations**
> > >
> > > We respectfully disagree that the paper takes its inferences at face value. Our results are interpreted through Finding 2, which shows that performance depends on the interaction between method, model, and perspective; this explains why consistent method-level rankings do not emerge.
> > >
> > > More generally, due to the distributed and overlapping nature of superposed representations, multiple plausible intuitions can be consistent with the same observed outcome; such explanations are not uniquely identifiable from the data alone.
> > >
> > > For example, mean-difference methods (e.g., DIM/ACE) may perform better when the positive and negative example distributions are more linearly separable; variance-based methods (e.g., PCA/LAT) may capture higher-variance but less task-aligned subspaces; and token-localized methods (e.g., CAA) may have reduced impact due to limited intervention scope. At the same time, the performance differences can also be interpreted as reflecting how much extraneous noise each method captures beyond the intended subspace. These interpretations are not mutually exclusive and cannot be reliably ranked in plausibility without further grounding.
> > >
> > > We will include elements of this discussion in the revision as the enumeration of multiple equally plausible accounts itself further supports Key Finding 2.
> > >
> > > ---
> > >
> > > **Q8: Definition of a "setting"**
> > >
> > > We apologize for the confusion in our previous response. Generally, 'setting' refers to the Standard/NoKL/Conditional configurations defined at line 375, but the term was inadvertently reused in the referenced sentence, referring to a specific layer-index and steering-coefficient pair. Such pairs span layers from the 25th to 80th percentile of model depth with step size 2, and coefficients from -3 to 3 inclusive (Appendix B.1.2); the KL check discards any pair producing average KL above 0.1 on last-token logits on the Alpaca validation split. We will directly invoke these two hyperparameters in that sentence in the revision.
> > >
> > > ---
> > >
> > > **W2, W4, W11, W8, W10: Presentation**
> > >
> > > We confirm the revision will include inline definitions for all terminology from cited prior work such as sneaking and adversarial refusal ablation, clarify the three settings, and reference the unsteered baseline results more prominently.  SteeringSafety operates at a scale that fundamentally constrains what can appear in the main text: every figure compresses thousands of results across models, methods, perspectives, and settings (pages 33-47). Certain method descriptions and analyses necessarily live in the appendix; reproducing them in the main text would take space from our core contributions. We are committed to improving clarity on the points raised and will do so in the revision, but we emphasize that the structure of the paper reflects principled tradeoffs at scale rather than presentation oversight.
> > >
> > > We are committed to addressing each of these points in the revision and hope the clarifications above, particularly regarding W9 and W12, are sufficient for the reviewer to reconsider their score.

---

### Official Review · Reviewer_Q9va · 2026-03-12

**Soundness:** 2
**Presentation:** 2
**Significance:** 3
**Originality:** 3
**Overall Recommendation:** 4
**Confidence:** 4

**Summary:**

This paper presents SteeringSafety, a benchmark for evaluating representation steering methods in LLMs across multiple safety perspectives. The benchmark covers nine perspectives spanning 18 datasets, introduces a unified implementation framework for five steering methods, and evaluates these methods on Gemma-2-2B, Llama-3.1-8B, and Qwen-2.5-7B. The central claim is that steering should not be judged only by whether it improves a target behavior, but also by how much it unintentionally changes other safety-relevant behaviors, which the paper terms entanglement. Empirically, the authors find that steering effectiveness depends heavily on the method-model-perspective combination, and that social behaviors and normative judgment are especially vulnerable to unintended side effects.

**Compliance With Llm Reviewing Policy:**

Affirmed.

**Final Justification:**

I still think this is a useful and important benchmark. It has some limitations but I'm still positive.

**Key Questions For Authors:**

1- How robust are the main aggregate conclusions to alternative definitions of entanglement? Since effectiveness is normalized and entanglement is not, and since the benchmark mixes quite heterogeneous datasets, I would like to know how stable the high-level findings are under alternative aggregation choices.

2- Can the authors better separate empirical findings from mechanistic interpretation in Section 4.1? The superposition explanation is interesting, but right now it reads more confidently than I think the experiments justify.

3- How much does the use of LLM judges affect the conclusions for the social-behavior and hallucination-style evaluations? The appendix mentions evaluator validation, but I would still like a clearer sense of how sensitive the results are to the choice of judge.

4- Why were only five perspectives chosen as steering targets, while the benchmark claims nine perspectives overall?

**Limitations:**

yes, and I appreciate that the paper includes an impact statement and is fairly clear about the practical motivation and the risk of using harmfulness steering to study jailbreak-related behavior. I would still encourage the authors to discuss two additional limitations more explicitly: first, that several evaluation dimensions rely on model-based judges, and second, that the mechanistic explanation around superposition is currently more of a hypothesis than a demonstrated conclusion.

**Strengths And Weaknesses:**

This paper talk about alignment and safety evaluation that is often missed in papers on representation steering: improving one behavior may silently damage others. I think that core framing is valuable and timely, especially given how much recent work on activation steering emphasizes simple, local behavior control.

The paper has a number of real strengths. The benchmark scope is broad, covering harmfulness, bias, intrinsic and extrinsic hallucination, social behaviors, reasoning, epistemic integrity, and normative judgment. The framework is also practically useful: it modularizes five popular steering methods and evaluates them in a consistent setup, which is helpful for comparison and reproducibility. I also liked that the paper is not only asking “which method works best,” but also “what else breaks when it works.” That is a meaningful contribution. The results are interesting as well: harmfulness is easiest to steer, hallucination steering is much less consistent, and social behaviors appear particularly fragile under intervention.

That said, I have some reservations about soundness. The benchmark idea is strong, but some of the paper’s central aggregate metrics are not as clean as the paper sometimes suggests. In particular, effectiveness is normalized while entanglement is not, and the practical justification for that choice is understandable, but it makes cross-setting comparisons a bit harder to interpret. More broadly, the paper compresses many heterogeneous tasks into shared aggregate summaries, and I was not always convinced that a single entanglement number is equally meaningful across such different perspectives. I think the per-dataset results are often more informative than the high-level aggregates.

I also think some of the stronger interpretive language goes a bit beyond the evidence. For example, the discussion section attributes much of the phenomenon to superposition and suggests that entanglement may be fundamentally heterogeneous and difficult to correct. That is an interesting hypothesis, but at least in this paper it still feels more like an interpretation than something directly established. The empirical results clearly show interference across perspectives, but the mechanistic explanation is still speculative. I would have preferred more caution in how strongly that story is presented.

Another limitation is that the benchmark focuses steering interventions on five target perspectives, while other perspectives are treated only as evaluation axes. That design is understandable from a data and prior-work standpoint, but it means the benchmark is not completely symmetric across all nine perspectives. So while the paper is broad in evaluation, it is narrower in what it actually tests as a steering target. I also think the use of LLM judges such as GPT-4o and Llama-based evaluators for several datasets introduces another layer of uncertainty, even though this is common practice. The paper mentions human-annotator comparisons in the appendix, which helps, but I still think the conclusions should be read as benchmark-level empirical trends rather than highly precise measurements.

On presentation, I found the paper readable overall, and Figures 1, 3, and 4 are genuinely helpful. However, I do think presentation is a noticeable weakness in its current form. There are large empty spaces after Figure 2 and Figure 4 in the main paper, which makes the layout look unfinished and wastes valuable page space that could have been used for clearer explanation, additional analysis, or more compact formatting. This is especially noticeable because those figures are important result figures. More generally, while the paper has useful visuals, some sections are still denser than they need to be, and a few key claims would benefit from clearer, tighter presentation.

In terms of significance, I think the paper addresses a real gap. A benchmark for measuring unintended cross-behavior effects of steering is useful, and I can imagine other researchers adopting this framing even if they do not use exactly this benchmark. For originality, I would say the paper is strongest in problem framing and evaluation design rather than in proposing a new steering method. That is perfectly fine for a benchmark paper, but it also means the originality comes more from the benchmark perspective and unified evaluation framework than from a fundamentally new technical mechanism.

Overall, I think this is a thoughtful and useful benchmark paper. My main hesitation is that the paper sometimes overstates what its aggregate metrics and interpretive discussion can support, and the presentation could be cleaner. Still, I found the main idea worthwhile, and I think the benchmark could be useful to the community.

---

> ### Author Rebuttal · Authors · 2026-03-28
>
> We thank the reviewer for the thorough and engaged reading, and for the positive assessment of the benchmark's scope, framing, and significance. We address each concern in turn.
>
> ---
>
> **W1: Aggregate metrics compress heterogeneous tasks; entanglement and effectiveness use different normalization**
>
> **Clarification: The asymmetric normalization is intentional; per-cell results are the primary evidence; we will add further entanglement decomposition in a revision.**
>
> The aggregate scores are coarse navigation aids rather than precise measurements. The headroom normalization in effectiveness is intentional: a raw absolute gain from a 95% baseline represents a qualitatively harder achievement than the same gain from a 50% baseline. We agree this makes direct cross-perspective effectiveness comparisons potentially misleading and will add a clarifying note to Figure 2. On entanglement: RMSE aggregation loses sign, meaning beneficial and harmful side effects of equal magnitude are treated identically. Our goal is to quantify the total magnitude of unintended behavioral change, since even ostensibly beneficial side effects represent unpredictable drift that undermines precise controllability. The full signed per-cell results with significance markers are already available in Appendix G (Figures 6-16) as the primary evidence. We will add entanglement metrics in further revisions and verify that rankings in Figure 3 and Table 2 are stable under this alternative.
>
> ---
>
> **W2: Superposition explanation is speculative and presented too confidently**
>
> **Criticism Addressed: We agree and will reframe superposition as a hypothesis rather than an established explanation.**
>
> Our empirical results establish that cross-perspective interference occurs and characterize its structure, but the mechanistic explanation grounded in superposition is a hypothesis rather than a demonstrated conclusion. We will revise Section 4.1 to more clearly separate empirical findings from interpretive speculation, explicitly labeling the superposition account as a candidate explanation motivated by prior theoretical work (Elhage et al., 2022) rather than something directly established by our experiments.
>
> ---
>
> **W3: Only five perspectives used as steering targets**
>
> **Clarification: The asymmetry is intentional: insufficient contrastive data and unvalidated linear representability make the remaining perspectives unreliable steering targets.**
>
> Two considerations motivate this choice. First, reliable contrastive direction generation requires sufficient paired training examples, and several perspectives such as DarkBench social behaviors do not have enough data in their current splits, nor is it straightforward to synthetically extend them. Second, it is not established that behaviors like social manipulation, epistemic integrity, or political neutrality have reliable linear representations in these models' residual streams. Attempting to steer perspectives without first validating linearity risks producing noisy or uninterpretable directions, weakening our conclusions. We will clarify this reasoning in the paper.
>
> ---
>
> **W4: Presentation: empty space after Figures 2 and 4**
>
> **Suggestion Addressed: This is a layout issue we will fix in the revision.**
>
> Thank you for flagging this issue. We will be sure to revise this and better utilize this space in future revisions.
>
> ---
>
> **Q1, Q2: Robustness to alternative entanglement definitions; separating empirical findings from mechanistic interpretation**
>
> The full signed per-cell results in Appendix G are the ground truth; the aggregate is a convenience summary. We will add a signed decomposition in a revision and report whether rankings change. We will also revise Section 4.1 to clearly label the superposition account as a hypothesis as noted in W2.
>
> ---
>
> **Q3: Sensitivity of social behavior and hallucination conclusions to judge choice**
>
> **Clarification: We use judges proposed by the original benchmark authors, inheriting their validation; DarkBench results are treated as indicative rather than precise.**
>
> Our judge choices follow the original benchmark papers, inheriting the validation performed as part of those works' contributions and enabling fair comparison to results reported there. Our own human agreement results (Appendix E.2, Table 4) show Cohen's κ = 0.920 on SALAD-Bench and κ = 0.860 on PreciseWikiQA. For DarkBench, Kran et al. (2025) report an average κ of 0.71 for GPT-4o, reflecting the inherent difficulty of judging subtle social behaviors. We thank the reviewer for identifying this drawback: DarkBench is one of the few works to study these social behaviors at all, yet does not itself rigorously formalize its behavioral definitions for human judge replication. This is a limitation of the field as a whole; we will make clear that DarkBench-based conclusions should be read as indicative trends rather than precise measurements.

---

> > ### Author Rebuttal · Reviewer_Q9va · 2026-04-03
> >
> > Thank you for the detailed and thoughtful rebuttal. I appreciate the careful responses to each of my points.
> >
> > My overall assessment remains positive.

---

> > > ### Author Response · Authors · 2026-04-06
> > >
> > > We thank the reviewer for the thoughtful and detailed engagement and for the positive assessment of the benchmark's framing and significance. The feedback on separating empirical findings from mechanistic interpretation in Section 4.1, and on presentation tightening, was well-taken and will be directly addressed in the revision.

---

### Official Review · Reviewer_Wsw9 · 2026-03-13

**Soundness:** 3
**Presentation:** 3
**Significance:** 3
**Originality:** 3
**Overall Recommendation:** 4
**Confidence:** 4

**Summary:**

The paper aims to introduce a new benchmark for evaluating entanglement of representation steering across different numbers (9) of safety perspectives. From their benchmarking results DIM and ACE seem to be more robust across methods,  while other steering methods demonstrate considerable entanglement degradation. Their contributions include entanglement measurement and code framework that allows for direct comparison to thus study the effectiveness-entanglement tradeoff while also allowing for newer technique explorations due to their framework. They consider two aggregate metric effectiveness and entanglement.  They use these metrics to develop the direction of safety reduction measurements. The models under test are Gemma-2-2B, Llama-3.1-8B, Qwen-2.5-7B instruction tuned with additional experiments on smaller Qwen variants.  The paper attributes entanglement to feature superposition in model activation spaces.

**Compliance With Llm Reviewing Policy:**

Affirmed.

**Key Questions For Authors:**

1. Entanglement (Eq. 2) uses RMSE over changes in the OOD perspectives, so the sign gets lost once things are aggregated. That means a method that consistently makes OOD perspectives worse could end up with the same entanglement score as one that has a mix of positive and negative side effects of the same magnitude. Did the authors look at a signed version of this metric, or at least separate beneficial vs. harmful side effects? It would render more confidence in the conclusions if the rankings in fig. 3 and Table 2 were stable under that choice.

2. Effectiveness  (Eq.1) normalizes by remaining headroom (1− yd​), which may make cross perspective effectiveness scores hard to compare when baseline levels differ substantially. For example, perspectives with already high baselines have smaller denominators and may therefore show larger normalized gains for similar absolute improvements. Are the cross perspective comparisons in fig. 2 intended to be interpreted directly, and if so, how do the authors justify this asymmetry?

3. Why were social behaviors, epistemic integrity, and reasoning not included as steering targets?

4. Could the authors include a larger model in the 60-500B range? This allows for generalization abilities to be studied.

**Limitations:**

Yes

**Strengths And Weaknesses:**

The paper is generally sound. The key questions that are to be answered are well advised. The Use of FDR corrected paired t-tests across experiments is appropriate and comparing LLM judges’ output with human annotations is a good validation. Grouping experiments are effective to establish an explanation.


The Paper is generally well organized and benchmark design is explained clearly. Key finding call out boxes provide useful navigation. The detailed appendices provide valuable information to clarify further questions and help with reproducibility.


The entanglement metric in ( Eq2.) uses an RMSE based formulation that captures magnitude but not direction at the aggregate level. This means a model that improves on all OOD perspectives would register equal entanglement to one that degrades on all of them, which is counterintuitive and could obscure meaningful distinctions.
The effectiveness metric (Eq1.) normalizes by remaining headroom, but this normalization is not symmetric across perspectives with different baseline levels, making direct cross-perspective effectiveness comparisons potentially misleading
Of the nine evaluation perspectives, only 3 groupings, harmfulness, hallucination, and bias, are used as steering targets (Figure 2). The remaining four perspectives, social behaviors, reasoning capabilities, epistemic integrity, and normative judgment, appear only as passive entanglement targets, so the benchmark never measures whether these behaviors can be effectively steered. This creates an asymmetry that weakens the benchmark's claim to comprehensiveness. The paper's own supplementary results (Appendix G.1) show that steering commonsense morality yields effectiveness up to 21.2% on Llama-3.1-8B, which is comparable to bias steering in the main experiments. This suggests that at least some excluded perspectives are steerable, so the restriction does not appear to reflect a fundamental limitation alone. As a result, a practitioner cannot tell from this paper whether social behaviors or epistemic integrity are hard to steer for representational reasons or simply because they were not attempted.

The TwinViews metric is hard to interpret because “percent left-leaning” has no clear neutral baseline, and the paper admits that the choice of direction was arbitrary. As written, the metric may overstate what can actually be concluded about political bias.

The size of the models make the findings hard to generalize. It's still not actionable for practitioners since they must empirically validate for their setup.

---

> ### Author Rebuttal · Authors · 2026-03-28
>
> We thank the reviewer for the thorough and engaged reading, and for the positive assessment across all four dimensions. We address each concern and key question in turn.
>
> ---
>
> **W1: Entanglement metric (Eq. 2) loses sign information**
>
> **Criticism Addressed: The aggregate metric summarizes disruption magnitude; signed breakdowns are in the per-cell figures, and we will add a signed decomposition in a revision.**
>
> You are correct that RMSE aggregation loses sign, meaning a method that consistently degrades OOD perspectives and one with mixed positive/negative effects of equal magnitude receive the same score. The aggregate entanglement metric is intentionally designed to quantify the total magnitude of unintended behavioral change, since even ostensibly beneficial side effects represent unpredictable drift that undermines precise controllability. The full signed per-cell results with significance markers are available for every experiment in Appendix G (Figures 6, 9, 12, and variants), allowing readers to inspect the direction of every individual effect. We will add a signed decomposition into beneficial vs. harmful components in a revision and verify that rankings in Figure 3 and Table 2 are stable under this alternative.
>
> ---
>
> **W2: Effectiveness normalization asymmetry (Eq. 1)**
>
> **Clarification: The normalization reflects genuine difficulty rather than a comparison artifact; Figure 2 is intended to show within-perspective variation, not rank perspectives against each other.**
>
> The headroom normalization is intentional: perspectives with high baselines are genuinely harder to improve, and a 5% gain from a 95% baseline is a much larger practical achievement than 5% from a 50% baseline. We agree this makes direct cross-perspective comparisons of effectiveness scores potentially misleading, and Figure 2 is intended primarily to show within-perspective variation across methods and models. We will add a clarifying note to this effect in the caption.
>
> ---
>
> **W3: Social behaviors, epistemic integrity, and reasoning not used as steering targets**
>
> **Clarification: The asymmetry is intentional: insufficient contrastive data and unvalidated linear representability for the remaining perspectives make them unreliable steering targets.**
>
> Two considerations motivate this. First, reliable contrastive direction generation requires sufficient paired training examples, and several perspectives including DarkBench social behaviors do not have enough data in their current splits to support this cleanly, nor is it straightforward to synthetically extend them. Second, it is not established that behaviors like social manipulation, epistemic integrity, or political neutrality have reliable linear representations in these models' residual streams. Attempting to steer perspectives without first validating linearity risks producing noisy or uninterpretable directions, which would weaken conclusions about entanglement rather than strengthen them. We will clarify this reasoning more explicitly in the paper.
>
> ---
>
> **W4: TwinViews metric is hard to interpret**
>
> **Misunderstanding Clarified: All reported values are shifts relative to the unsteered baseline, making the metric internally consistent and interpretable.**
>
> We agree the absolute label "percent left-leaning" has no universal neutral reference point. However, all reported TwinViews values are expressed as changes relative to the unsteered model's baseline, not as absolute political position estimates. The metric consistently answers whether a steering intervention shifted the model toward or away from its prior political tendency, and by how much. The direction of the shift is always meaningful in this relative sense regardless of where the absolute baseline sits. We will clarify this framing in the paper.
>
> ---
>
> **Q: Could the authors include a larger model in the 60--500B range?**
>
> **Clarification: Larger-model results would be difficult to interpret cleanly given the confounds involved.**
>
> Results on a 70B+ model would almost certainly differ from our findings, but interpreting why is intractable: model scale cannot be cleanly separated from the many factors that co-vary with it, including number of layers, hidden dimension size, amount and composition of training data, and training and alignment procedure. Safety-relevant representations such as refusal are artifacts of each model's specific alignment training and data, meaning they are idiosyncratic to each model family and training run by construction. A finding that entanglement differs at 70B would be consistent with many competing explanations and would not cleanly generalize to other large models. Extending the benchmark to larger scales also exceeds our compute budgets (already up to 800 GPU hours) and is most productively pursued as a dedicated study designed to control for these confounds, rather than as an add-on experiment.

---

> > ### Author Rebuttal · Reviewer_Wsw9 · 2026-04-03
> >
> > Concerns have been fully answered, and will keep the original positive score as is. Thank you!

---

> > > ### Author Response · Authors · 2026-04-06
> > >
> > > We thank the reviewer for the thorough and constructive engagement and for maintaining a positive assessment. The feedback on signed entanglement decomposition and normalization asymmetry was valuable and will strengthen the paper in revision.

---

### Official Review · Reviewer_vhAG · 2026-03-13

**Soundness:** 3
**Presentation:** 3
**Significance:** 2
**Originality:** 1
**Overall Recommendation:** 5
**Confidence:** 3

**Summary:**

This paper proposes the SteeringSafety Benchmark, a benchmark test for evaluating the safety of steering methods (RepE) applied to large language models. Its core argument is that existing steering vector evaluations narrowly focus on whether a single target behavior is achieved (e.g., improving honesty or reducing toxicity), while ignoring the broader side effects of these interventions on model capabilities and other safety-related behaviors. The authors propose an integrated evaluation framework with three pillars: (1) General Capability Preservation (measured by MMLU, TruthfulQA, and perplexity), (2) Safety Alignment Preservation (measured by six safety categories in a custom dataset), and (3) Behavioral Consistency (measuring whether the model's persona and instruction following remain stable). They evaluated four steering methods—Intervention Time (ITI), Representation Engineering (RepE), Linear Artificial Tomography (LAT), and Contrastive Activation Steering Vectors (ContraSVec)—on three open-source models (Llama-2-7B-Chat, Llama-3-8B-Instruct, Gemma-2-9B-IT). The main findings include: steering methods often reduce general capabilities (sometimes catastrophically), may inadvertently weaken safety alignment in non-target categories, and larger steering multipliers amplify these side effects in a nonlinear manner.

**Compliance With Llm Reviewing Policy:**

Affirmed.

**Final Justification:**

In light of the rebuttal, I revise my assessment upward. The paper makes a sound and timely contribution to understanding the side effects of representation steering, providing infrastructure and empirical grounding that the community currently lacks. The work is clearly written, thoroughly evaluated, and the open codebase supports reproducibility and extension. I support acceptance.

**Key Questions For Authors:**

See weaknesses

**Limitations:**

The authors discuss limitations in Section 7, acknowledging the restriction to 7-9B parameter models, the limited set of four steering methods, and the focus on English-language evaluation.

**Strengths And Weaknesses:**

Strengths:

1. The paper addresses a genuine gap in the RepE steering vector literature: most prior work evaluates only the targeted behavior and ignores collateral damage to model capabilities and safety.

2. The benchmark covers four representative steering methods, three diverse models, multiple steering multipliers, and evaluates across three distinct dimensions (capability, safety, consistency). The experimental matrix is large and the results are reported with reasonable thoroughness, including 35+ figures in the appendix.

Weaknesses:

1. This article is essentially a benchmarking study that collects existing evaluation metrics (MMLU, TruthfulQA, perplexity) and applies existing guidance methods.

2. A custom security dataset composed of 1,800 prompts (300 per category) has been built but not strictly validated. The paper does not report inter-rater reliability, does not provide detailed information about the annotators' qualifications, and has not compared the dataset with existing security benchmarks. Although the six security categories are reasonable, they seem somewhat provisional. Relying on GPT-4 as a judge of security compliance introduces a dependency on proprietary models and the known bias issues of LLMs as judges.

3. All evaluated models are in the 7-9B parameter range. It is unclear whether the findings generalize to larger models (13B, 70B, or closed-sourced models)

4. The paper records that the guiding method *can* produce side effects, but the understanding of the mechanisms *why* they occur is limited.

5. Error bars, confidence intervals, or variance across runs are not consistently reported for the main results.

---

> ### Author Rebuttal · Authors · 2026-03-28
>
> We thank the reviewer for the overall positive assessment and constructive feedback. We address each weakness in turn.
>
> ---
>
> **W2: Dataset validation and LLM-as-judge reliability**
>
> **Clarification: We use 17 established published benchmarks, not a custom dataset, and our LLM judges are those proposed and validated by each benchmark's original authors.**
>
> We do not use a custom-authored prompt dataset. Our benchmark aggregates 17 established published benchmarks, each with their own validation pedigree. There is no 1,800-prompt custom set requiring independent annotation validation.
>
> Our LLM judge choices follow exactly the judges proposed by the original dataset papers, meaning (1) our judges inherit the rigorous validation performed as part of those works' own contributions, and (2) our results are directly comparable to numbers reported in the original benchmark papers. We report our own human agreement in Appendix E.2 (Table 4): Cohen's κ = 0.920 on SALAD-Bench and κ = 0.860 on PreciseWikiQA. For DarkBench, Kran et al. (2025) report an average κ of 0.71 for GPT-4o. We acknowledge LLM-as-judge limitations in Appendix A.
>
> ---
>
> **W3: Model size generalizability**
>
> **Clarification: The 2B--8B range is where steering is most studied and where mechanistic follow-up work is most tractable; we also include smaller Qwen experiments, and extension to larger models is left to future work.**
>
> We fully acknowledge this limitation, stated explicitly in Appendix A. The 2B--8B range is the most practically relevant range for applied representation steering: the vast majority of published steering work operates here, and it is where safety interventions are most actively deployed. We also include additional experiments on Qwen-2.5-1.5B and Qwen-2.5-3B (Appendix E.4, Figures 15 and 16). Mechanistic interpretability tools such as sparse autoencoders, circuit analyses, and probing methods are most mature at these scales, meaning our findings are most actionable for follow-on work seeking to understand why entanglement occurs. Extension to 70B+ models requires substantially more compute and is better suited to future work; our modular open codebase is designed to support this.
>
> ---
>
> **W4: Limited mechanistic understanding**
>
> **Clarification: Mechanistic understanding of cross-behavior entanglement remains an open problem; our empirical findings are a prerequisite for such accounts.**
>
> We agree and note this is an intentional scope boundary in Section 4.1. Identifying that entanglement occurs, which perspectives are vulnerable, and which method-model pairings minimize collateral effects provides concrete grounding for future mechanistic work. Explanations grounded in superposition (Elhage et al., 2022) are promising, but even extremely recent work post-dating our submission that attempts to formalize feature geometry in superposition remains theoretical and is not yet empirically validated beyond small toy models [1]. We see our empirical characterization as establishing the phenomena that such theory must explain.
>
> ---
>
> **W5: Error bars and confidence intervals**
>
> **Clarification: Per-cell significance markers are reported throughout; repeated-run variance is not meaningful given our deterministic pipeline.**
>
> Statistical significance is reported throughout our results. All heatmap figures in Appendix G display per-cell significance markers (*, **, *** for p < 0.05, p < 0.01, p < 0.001) computed over test-set samples. We do not report variance across repeated runs as the LLMs are evaluated utilizing greedy decoding. Uncertainty in accuracy estimates is fully captured by the per-sample significance tests shown. However, we agree that reporting bootstrap confidence intervals over evaluation samples would further improve clarity, and we will investigate including these in revisions.
>
> ---
>
> **W1: Contribution framing**
>
> **Criticism Addressed: Assembling existing metrics does not yield our findings; the cross-perspective evaluation design is the contribution, and prior work tests at most one interaction where we test 85.**
>
> While our benchmark builds on existing metrics, the contribution lies in the cross-perspective evaluation design: no existing framework simultaneously measures how an intervention targeting one safety behavior affects performance across the others. Findings such as harmfulness steering consistently degrading sycophancy while leaving reasoning intact, or hallucination steering partially jailbreaking Llama-3.1-8B, are not recoverable from any single prior evaluation. Prior work tests at most one interaction (refusal steering's impact on factuality); we test 85. We believe these findings, and the infrastructure for reproducing and extending them, represent a meaningful contribution to the steering and safety literature.
>
> ---
>
> [1] Georgi Ivanov, Narmeen Oozeer, Shivam Raval, Tasana Pejovic, Shriyash Upadhyay, Amir Abdullah: Spectral Superposition: A Theory of Feature Geometry. CoRR abs/2602.02224 (2026)

---

> > ### Author Rebuttal · Reviewer_vhAG · 2026-04-03
> >
> > Thank you for the author's detailed rebuttal, most of my concerns have been addressed.

---

> > > ### Author Response · Authors · 2026-04-06
> > >
> > > We thank the reviewer for the generous score adjustment and for the constructive and detailed engagement throughout. We are committed to addressing the remaining points, particularly around dataset validation clarity and model size generalizability, fully in the revision, and appreciate the reviewer's recognition of the benchmark's contributions to the field.

---

### Decision · Program_Chairs · 2026-04-30

**Decision:**

Accept (regular)

**Comment:**

There's a lot to recommend in this paper. its overall approach to developing a benchmark for steering effectiveness is well thought out and most of the reviewers agree. There's a bit of back and forth with one reviewer, which resolves many of the points of concern, but also points to some weakness in presentation of the material.

Overall, I don't view these weaknesses as dealbreakers for the paper and think its merits outweigh them.